# Tight Bounds for Learning RUMs from Small Slates

**Flavio Chierichetti**
Sapienza University of Rome
`flavio@di.uniroma1.it`

**Mirko Giacchini**
Sapienza University of Rome
`giacchini@di.uniroma1.it`

**Ravi Kumar**
Google, Mountain View
`ravi.k53@gmail.com`

**Alessandro Panconesi**
Sapienza University of Rome
`ale@di.uniroma1.it`

**Andrew Tomkins**
Google, Mountain View
`atomkins@gmail.com`

## Abstract

A Random Utility Model (RUM) is a classical model of user behavior defined by a distribution over $\mathbb{R}^n$. A user, presented with a subset of $\{1, \ldots, n\}$, will select the item of the subset with the highest utility, according to a utility vector drawn from the specified distribution. In practical settings, the subset is often of small size, as in the "ten blue links" of web search.

In this paper, we consider a learning setting with complete information on user choices from subsets of size at most $k$. We show that $k = \Theta(\sqrt{n})$ is both necessary and sufficient to predict the distribution of all user choices with an arbitrarily small, constant error.

Based on the upper bound, we obtain new algorithms for approximate RUM learning and variations thereof. Furthermore, we employ our lower bound for approximate RUM learning to derive lower bounds to fractional extensions of the well-studied $k$-deck and trace reconstruction problems.

## 1 Introduction

In many common settings, both online and offline, users select from a set of available candidates: cars on a dealer's lot; songs on a streaming service; movies in a Netflix carousel of choices; and so forth. Often, it is unrealistic to offer the user the entire universe of items. No car dealership has every new and used car ever produced. Likewise, recommendation services have enormous catalogs of songs, products, movies, etc, and must carefully curate a more manageable subset of recommended items that will fit within the constraints of the user interface. Thus, user feedback often arrives as a choice from slates of items of a certain standard cardinality—think of "ten blue links" in web search as the classical example.

Random Utility Models (commonly called *RUMs*) have been the standard mathematical model for studying user choices over subsets of a universe of items. RUMs are the subject of many decades of study, and the centerpiece of the 2000 Nobel prize in economics. The model family is straightforward: a RUM operates over a universe $U$ of items, with $|U| = n$, and is characterized by a distribution over $\mathbb{R}^n$. A user draws from this distribution to generate a vector specifying the utility of each item of $U$. The user is offered a subset of $U$ called a *slate*, and must select a single item from the slate; the user behaves rationally by selecting the available item of the highest utility. The *winning distribution* of the slate is a probability distribution over the items of the slate representing the likelihood (over draws from the utility distribution) that a particular item is selected. To learn a RUM, an algorithm is given a training set of examples of slates with their winning distributions, and must then guess the winning distributions of a new test set of slates.

38th Conference on Neural Information Processing Systems (NeurIPS 2024).

We are interested in learning RUMs when the example slates in the training set are constrained to have at most a certain cardinality. We will then deem an algorithm successful if exposure to these smaller slates allows it to approximate the winning distribution of all slates. In the extreme example, the algorithm should infer the winning distribution over the universal slate $U$ itself, representing the likelihood that a particular item is a random user's favorite from the entire catalog.

**Our contributions.** We present two main results representing paired upper and lower bounds for this question. The upper bound shows that, with knowledge of the winning distributions for slates of size at most $O(\sqrt{n})$, one can approximate the winning probability for any item in any slate to within an arbitrarily small additive constant. Using this upper bound, we obtain an exponential improvement in the time to learn a RUM. The previous best-known algorithm, implicit in earlier work, requires time $2^{O(n)}$. Our new algorithm learns any RUM to within an $\ell_\infty$-error (resp., $\ell_1$-error) of $\epsilon > 0$ in time $n^{O\left(\sqrt{n \log \frac{1}{\epsilon}}\right)}$ (resp., $n^{O\left(\sqrt{n \log \frac{n}{\epsilon}}\right)}$). We also give a "simulation" result: the winning distribution of each slate $T$ can be approximated within a constant $\ell_\infty$-error by querying polynomially many sub-slates of $T$, each of size at most $O\left(\frac{|T|}{\log |T|}\right)$.

Our near-matching lower bound shows that, with knowledge of the winning distributions for all slates up to size $o(\sqrt{n})$, any algorithm must make error $1/2 - \epsilon$ (almost the worst possible) in predicting a target item's probability in the universal slate, for arbitrarily small constant $\epsilon$. Based on this lower bound, we also obtain lower bounds to fractional extensions of the well-studied $k$-deck and trace reconstruction problems.

The bounds for RUM learning algorithms depend on the nature of the oracle used to present examples to the algorithm. For a given slate, the MAX-DIST oracle returns the exact winning distribution, while the MAX-SAMPLE oracle simply returns a draw from the winning distribution, as we would expect from a real-world setting. The algorithms we present work with both oracles, with asymptotically the same running time and the same sample complexity.

**Overview of techniques.** Our results are proved by observing a series of connections between RUM learning and the approximation of the bitwise AND function via polynomials. Specifically, the two main quantities of interest in approximating Boolean functions with polynomials are (i) the degree of the polynomial, and (ii) the $\ell_1$-norm of the coefficients of the polynomial. We will show that these two quantities are related, respectively, to (i) the size of the slates required to approximate a RUM (Theorem 9 and Theorem 12), and (ii) the running time to approximate the winning distribution of a queried slate (Theorem 9 and Corollary 10).

**Related work.** Discrete choice has been the subject of extensive research in machine learning and economics; see [Train, 2003] for an excellent introduction. RUMs are an important class of models in discrete choice—in particular, Multinomial Logits (MNLs) and their mixtures are special classes of RUMs. RUMs have been extensively studied from both active and passive learning perspectives [Soufiani et al., 2012, Oh and Shah, 2014, Chierichetti et al., 2018a,b, Negahban et al., 2018, Tang, 2020] and from an efficient representation point of view [Farias et al., 2009, Chierichetti et al., 2021].

A number of papers have used linear programming (LP) for obtaining representation of RUMs that agree with an empirical distribution on small slates [Farias et al., 2009, Almanza et al., 2022, Chierichetti et al., 2023]. In particular, Farias et al. [2009] make strong assumptions on the underlying RUM, while in [Almanza et al., 2022, Chierichetti et al., 2023] the algorithms are not required to generalize outside of the observed training set. Instead, we are asking what is the minimum $k$ such that if one observes all the winning distributions of slates of size up to $k$, one can approximately reconstruct *all* the winning distributions of a RUM? Note that we do not make any assumption on the RUM and our algorithms must generalize to all the slates.

The approximation of Boolean functions via polynomials has found applications in a disparate and apparently remote number of fields such as cryptography [Bogdanov et al., 2016], differential privacy [Thaler et al., 2012], quantum query complexity [Beals et al., 2001], PAC learning [Klivans and Servedio, 2004], and more. Our work draws connections between this field and RUM learning for the first time, showing yet another important application of such techniques.

In this paper we also strengthen the relation, that was first observed in [Chierichetti et al., 2018a], between RUM learning and the $k$-deck problem.

**Organization.** In Section 2 we introduce the definitions and notation. In Section 3 we prove our upper bound, which we then use in Section 4 to obtain algorithms for learning RUMs. In Section 5, we present our lower bound. Finally, in Sections 6 and 7 we derive, as corollaries of our results, lower bounds for other learning problems. All the proofs missing from the main body of the paper can be found in Appendix A.

## 2 Background

Let $[n] = \{1, \ldots, n\}$ and let $\mathbf{S}_n$ denote the symmetric group on $n$ items (the set of all permutations of $[n]$). Let also $\binom{S}{k} = \{T \subseteq S \mid |T| = k\}$. For a distribution $D$, $x \sim D$ denotes that the random variable $x$ is drawn from $D$ and $D(i)$ denotes $\Pr_{x \sim D}[x = i]$, where $i \in \mathrm{supp}(D)$. For $a, b \in [n], \pi \in \mathbf{S}_n$ we write $a <_\pi b$ or $b >_\pi a$ to say that $\pi$ ranks $b$ higher than $a$.

**Random utility models.** A *slate* is a non-empty subset of $[n]$. For a slate $\varnothing \neq S \subseteq [n]$ and a permutation $\pi \in \mathbf{S}_n$, let $\pi(S)$ be the item of $S$ that wins, i.e., ranks the highest in $\pi$.

The following definition of RUMs, based on probability distributions over permutations, is equivalent to the utility-vectors definition given in the Introduction [Chierichetti et al., 2018a]; we adopt the distribution-over-permutations definition, since it makes it easier to present our algorithms.

**Definition 1** (Random Utility Model (RUM)). *A random utility model (RUM) $R$ on $[n]$ is a distribution on $\mathbf{S}_n$. For a slate $S$, $R_S$ denotes the distribution of the random variable $\pi(S)$ where $\pi \sim R$, so $R_S(s) = \Pr[s \text{ wins in } S]$ is called the* winning distribution *on $S$ induced by RUM $R$.*

We consider two types of oracle access to the RUM: the MAX-DIST oracle returns for a given slate the (exact) winning distribution for the slate and the MAX-SAMPLE oracle returns a draw from the winning distribution.

**Approximation of** AND. For a bit string $x = x_1 \ldots x_n \in \{0,1\}^n$ and $S \subseteq [n]$, let $\chi_S(x) = \prod_{i \in S} x_i$. By convention, $\chi_\varnothing(x) = 1$. The function $\mathrm{AND}_n : \{0,1\}^n \to \{0,1\}$ is the bitwise-AND given by $\mathrm{AND}_n(x) = \chi_{[n]}(x)$. We write only AND when $n$ is clear from the context. A polynomial $p : \{0,1\}^n \to \mathbb{R}$ is said to $\epsilon$-approximate the AND function if for all $x \in \{0,1\}^n$ it holds $|\mathrm{AND}(x) - p(x)| \leq \epsilon$. The $\epsilon$-*approximate degree* of AND is the smallest value $\deg_\epsilon(\mathrm{AND})$ such that there exists a polynomial of such degree that $\epsilon$-approximates the AND function, and it is well known that the optimal value is $\deg_\epsilon(\mathrm{AND}) = \Theta(\sqrt{n \log(1/\epsilon)})$ [Bun and Thaler, 2022]. The general form of a degree-$k$ polynomial is $\sum_{S \subseteq [n], |S| \leq k} a_S \cdot \chi_S(x)$, where the $\{a_S\}_{S \subseteq [n], |S| \leq k}$ are real coefficients. However, all the polynomials proposed in the literature have at most $k + 1$ distinct coefficients $a_0, a_1, \ldots, a_k$, such that $a_S = a_{|S|}$, therefore in this work we will focus only on polynomials of such form.[1] While it is folklore that such coefficients can be computed in polynomial time, for completeness we provide in Appendix D an explicit algorithm for this task. We will use several results on the approximation of the AND function that we introduce as needed.

## 3 Uniform approximation of a RUM

In this section we show that if two RUMs agree on small slates, then they nearly agree on all slates. Our result can be obtained as a consequence of a more general result of Sherstov [2008], which is based on the approximate degree of the AND function and was originally stated only for sub-constant errors. The following is a restatement of Sherstov [2008, Theorem 4.8], using the upper bound on the approximate degree of AND proved by [Buhrman et al., 1999]; we also make the error term explicit.

**Theorem 2** (Sherstov [2008]). *There exists a constant $c > 0$ such that the following holds. Consider any two probability spaces $\mathcal{P}_1$ and $\mathcal{P}_2$, and any events $A_1, \ldots, A_n$ in $\mathcal{P}_1$ and $B_1, \ldots, B_n$ in $\mathcal{P}_2$ such that, for any $S \subseteq [n], |S| \leq c \cdot \sqrt{n \cdot \ln 1/\epsilon}$, it holds $\Pr_{\mathcal{P}_1}[\cap_{i \in S} A_i] = \Pr_{\mathcal{P}_2}[\cap_{i \in S} B_i]$. Then, it holds: $\left| \Pr_{\mathcal{P}_1}[\cap_{i \in [n]} A_i] - \Pr_{\mathcal{P}_2}[\cap_{i \in [n]} B_i] \right| \leq \epsilon$, where $\epsilon \in (2^{-n}, 1/3)$*

---

[1]In the literature, other works focus on the coefficients of univariate polynomials $q : [0, n] \to \mathbb{R}$ taking in input the number of bits set to one. We stress that this is not the case in this work, where we always consider coefficients of multivariate polynomials $p : \{0,1\}^n \to \mathbb{R}$ unless otherwise specified.

**Theorem 3** (RUMs Upper Bound). *Let $P$ and $Q$ be two RUMs on $[n]$. There exists a constant $c > 0$ such that for a given $s \in [n]$, $T \subseteq [n] \smallsetminus \{s\}$, if $P_{S\cup\{s\}}(s) = Q_{S\cup\{s\}}(s)$ for each $S \in \left\{ T' \mid T' \subseteq T \text{ and } |T'| \le c \cdot \sqrt{|T| \cdot \ln \frac{1}{\epsilon}} \right\}$, then $\left| P_{T\cup\{s\}}(s) - Q_{T\cup\{s\}}(s) \right| \le \epsilon$, where $\epsilon \in (2^{-|T|}, 1/3)$.*

*Proof.* Let $c$ be the constant of Theorem 2 and define $k = c \cdot \sqrt{|T| \cdot \ln 1/\epsilon}$. Consider the probability space $\mathcal{P}_1$ (resp. $\mathcal{P}_2$) having $\mathbf{S}_n$ as sample space and RUM $P$ (resp. $Q$) as the probability mass function. For $t \in T$, let $A_t$ be the event $\{\pi \in \mathbf{S}_n \mid s >_\pi t\}$. Then, for any $S \subseteq T$, $\{\pi \in \mathbf{S}_n \mid \pi(S \cup \{s\}) = s\} = \cap_{i \in S} A_i$. Therefore, for any $S \subseteq T, |S| \le k$ it holds:

$$\Pr_{\mathcal{P}_1}[\cap_{i \in S} A_i] = \Pr_{\pi \sim P}[\pi(S \cup \{s\}) = s] = P_{S\cup\{s\}}(s) = Q_{S\cup\{s\}}(s) = \Pr_{\mathcal{P}_2}[\cap_{i \in S} A_i],$$

where the third equality follows by hypothesis. Finally, by Theorem 2:

$$\left| P_{T\cup\{s\}}(s) - Q_{T\cup\{s\}}(s) \right| = \left| \Pr_{\mathcal{P}_1}[\cap_{i \in T} A_i] - \Pr_{\mathcal{P}_2}[\cap_{i \in T} A_i] \right| \le \epsilon. \qquad \square$$

In light of Theorem 3, accessing slates of size up to $O\left(\sqrt{n \cdot \ln \frac{1}{\epsilon}}\right)$ is enough to predict the winning distribution of all the slates, within an additive $\epsilon$. In the next section, we obtain a computational version of this result.

## 4 Reconstruction algorithms

In this section we obtain two algorithms for reconstructing the winning distributions on large slates using the winning distribution on small slates. The goal of these algorithms is to obtain a computational version of Theorem 3. The first algorithm is a proper learning algorithm that outputs a RUM. Building the RUM takes time $n^{O(n)}$ but once built, querying this RUM to get the approximate winning distribution of any given slate takes only polynomial time. Moreover, using previous work [Chierichetti et al., 2021], this RUM can actually be approximately represented using $O(n^2 \log n)$ bits. The second algorithm is an improper learning algorithm: while its output model allows uniformly approximating the winning distribution on each slate, this model might not be a RUM. Building the model takes time $n^{O(\sqrt{n})}$ and once built, querying this model to get the approximate winning distribution of any given slate takes time $2^{O(\sqrt{n})}$. The total bit complexity of the second algorithm's model is $n^{O(\sqrt{n})}$. This second algorithm has two nice properties: (i) if we are given access to slates larger than $\sqrt{n}$, then querying the model becomes more efficient, and (ii) if we want to estimate the winning distribution of only $M = \mathrm{poly}(n)$ pre-determined slates, then building the model becomes more efficient. Putting these two properties together we are able to prove a "simulation" result: for any pre-determined slate $T \subseteq [n]$ it is possible to estimate $R_T$ to within a constant $\ell_\infty$-error $\epsilon$ in polynomial time and accessing slates of size at most $O\left(\frac{|T|}{\ln |T|}\right)$.

### 4.1 A proper learning algorithm

Fix a large enough integer $t \le n - 1$. For a RUM $Q$, let the winning distributions of slates of size at most $k$, for $k = O\left(\sqrt{t \ln \frac{1}{\epsilon}}\right)$ be known. To estimate the probability distributions $Q_T$ for any $s \in [n]$ and for any slate $T \subseteq [n] \smallsetminus \{s\}$ such that $|T| \le t$, it is sufficient to solve the following linear program (LP), with no objective function:

$$\begin{cases} \sum_{\substack{\pi \in \mathbf{S}_n \\ \pi(S\cup\{s\})=s}} p_\pi = Q_{S\cup\{s\}}(s) & \forall s \in [n] \; \forall S \subseteq [n] \smallsetminus \{s\} \text{ s.t. } |S| \le k-1 \\ \sum_{\pi \in \mathbf{S}_n} p_\pi = 1 \\ p_\pi \ge 0 & \forall \pi \in \mathbf{S}_n \end{cases} \tag{1}$$

Indeed, (1) returns a RUM $P$ that is compatible with RUM $Q$ on each slate of size at most $k$. Applying Theorem 3, we obtain:

**Observation 4.** *For $k = \Theta\left(\sqrt{t \ln \frac{1}{\epsilon}}\right)$, let $P$ be the RUM obtained by solving (1). Then, for any $s \in [n]$, and for any $T \subseteq [n] \smallsetminus \{s\}$ such that $|T| \leq t$, it holds $|P_{T \cup \{s\}}(s) - Q_{T \cup \{s\}}(s)| \leq \epsilon$*

By fixing $t = n - 1$ and solving (1)—an LP with $n!$ variables and $n^{O\left(\sqrt{n \ln \frac{1}{\epsilon}}\right)}$ constraints—we get:

**Theorem 5** (Proper learning algorithm)**.** *Let $Q$ be a RUM over $[n]$. There exists an algorithm that uses the* MAX-DIST *oracle on each slate of size at most $O\left(\sqrt{n \ln \frac{1}{\epsilon}}\right)$ and in time $n^{O(n)}$ produces a RUM $P$ such that for each $S \subseteq [n]$ and for each $i \in S$, $|P_S(i) - Q_S(i)| \leq \epsilon$.*

Using the result in [Chierichetti et al., 2021], the RUM $P$ can be subsampled in $\mathrm{poly}(n)$ time to a uniform RUM $\widetilde{P}$ with a multiset of $O\left(n/\epsilon^2\right)$ permutations as its support, and such that for each $S \subseteq [n]$, $\left|\widetilde{P}_S - P_S\right|_1 \leq \epsilon$. Thus, by accessing slates of size at most $O\left(\sqrt{n \ln \frac{1}{\epsilon}}\right)$ (resp., $O\left(\sqrt{n \ln \frac{n}{\epsilon}}\right)$), one can produce a data structure $\widetilde{P}$ in time $n^{O(n)}$ such that (i) $\widetilde{P}$ can be represented with $O\left(\epsilon^{-2} \cdot n^2 \log n\right)$ bits, and (ii) when $\widetilde{P}$ is queried on a slate $S$, it can return in $\mathrm{poly}(n)$ time a distribution $\widetilde{P}_S$ such that $\left|\widetilde{P}_S - Q_S\right|_\infty \leq \epsilon$ (resp., $\left|\widetilde{P}_S - Q_S\right|_1 \leq \epsilon$).

By providing a version of Theorem 3 that holds when the small slates are approximately equal, this algorithm can also be made to work with MAX-SAMPLE oracle. More details are given in Appendix B.

We mention that this algorithm can be made to run in time $2^{O(n)}$ by using the ellipsoid method and the separation oracle of Chierichetti et al. [2023]; more details are given in Appendix C.

## 4.2 An improper learning algorithm

In this section we obtain a learning algorithm whose data structure is not a RUM but can be built faster. As before, this is a restatement of Sherstov [2008, Theorem 4.8]:

**Theorem 6** (Sherstov [2008])**.** *Consider any probability space $\mathcal{P}$, and any events $A_1, \ldots, A_n$ in $\mathcal{P}$. For $k \geq \Theta\left(\sqrt{n \cdot \ln 1/\epsilon}\right)$, $\epsilon \in (2^{-n}, 1/3)$, let $\{a_i\}_{0 \leq i \leq k}$ be the coefficients of a degree $k$ polynomial approximating the* AND *function within $\epsilon$ (for any $S \subseteq [n], |S| \leq k$, the coefficient of the monomial $\chi_S(x)$ is $a_{|S|}$). Then, $\left|\mathrm{Pr}_{\mathcal{P}}\left[\cap_{i \in [n]} A_i\right] - \sum_{S \subseteq [n], |S| \leq k} a_{|S|} \cdot \mathrm{Pr}_{\mathcal{P}}\left[\cap_{i \in S} A_i\right]\right| \leq \epsilon$*

Given a RUM $R$ over $[n]$ and access to slates of size $k \geq \Theta\left(\sqrt{|T| \log \frac{1}{\epsilon}}\right)$, consider an element $s \in [n]$ and a slate $T \subseteq [n] \smallsetminus \{s\}$. Let $\{a_i\}_{0 \leq i \leq k-1}$ be the coefficients of a polynomial of degree $k - 1$ that approximates the $\mathrm{AND}_{|T|}$ function. Then, the following is a good approximation for $R_{T \cup \{s\}}(s)$:

$$\widetilde{R}_{T \cup \{s\}}(s) = \sum_{S \subseteq T, |S| \leq k-1} a_{|S|} \cdot R_{S \cup \{s\}}(s).$$

In fact, choosing the probability space and events as in Theorem 3 and applying Theorem 6, we get:

**Observation 7.** $\left|\widetilde{R}_{T \cup \{s\}}(s) - R_{T \cup \{s\}}(s)\right| \leq \epsilon$, where $R$ is a RUM over $[n]$, $s \in [n]$, $T \subseteq [n] \smallsetminus \{s\}$, and for $k \geq \Theta\left(\sqrt{|T| \ln \frac{1}{\epsilon}}\right)$, $\{a_i\}_{0 \leq i \leq k-1}$ are the coefficients of a polynomial of degree $k - 1$ approximating the $\mathrm{AND}_{|T|}$ function (the coefficient of the monomial $\chi_S(x)$ is $a_{|S|}$).

From the above observation and given MAX-DIST oracle access to slates of size $k = \Theta(\sqrt{n \ln(1/\epsilon)})$, we obtain a deterministic algorithm that first stores $R_S(s)$ for all $s \in S \subseteq [n], |S| \leq k$, in time $n^{O(k)}$ and then, upon query $(s, T)$ returns the approximation $\widetilde{R}_{T \cup \{s\}}(s)$ that can be computed in $|T|^{O(\sqrt{|T| \ln(1/\epsilon)})} \leq n^{O(k)}$ time[2].

Note that this result holds for any polynomial approximating the AND function. To get a better algorithm, which also works with MAX-SAMPLE oracle, we focus on a specific polynomial. (Observe

---

[2]We assume $|T| \geq k$ otherwise computing $R_{T \cup \{s\}}(s)$ is trivial since we can just query the oracle

also that the time to answer a query in the previous algorithm does not improve as $k$ increases. This is counter-intuitive: given access to larger slates, it should become easier to approximate the target slate. The second algorithm that we provide gets faster as $k$ increases, overcoming this limitation.)

The polynomial that we use is the one proposed by Huang and Viola [2022, Corollary 1.5]:

**Theorem 8** (Huang and Viola [2022]). *For all $\epsilon \in (2^{-n}, 1/3)$, $\sqrt{n\ln(1/\epsilon)} \le d \le n$, there exists a polynomial $p : \{0,1\}^n \to \mathbb{R}$ of degree $k = \Theta(d)$ and real coefficients $\{a_i\}_{0 \le i \le k}$, where the coefficient of $\chi_S(x)$ is $a_{|S|}$, such that: (i) for each $x \in \{0,1\}^n$, $|p(x) - \mathrm{AND}_n(x)| \le \epsilon$, and (ii) $\sum_{S \subseteq [n], |S| \le k} |a_{|S|}| = \sum_{c=0}^{k} \binom{n}{c} |a_c| \le 2^{O\left(\frac{n\ln(1/\epsilon)}{k}\right)}$.*

**Theorem 9** (Improper learning algorithm). *Let $R$ be a RUM over $[n]$. Let $d \ge \sqrt{n\ln(1/\epsilon)}$, $\epsilon \in (0, 1/3)$, and $\delta \in (0,1)$ such that $\epsilon, \delta \ge \frac{1}{n^{O(1)}}$. Then, there exists a randomized algorithm that accesses slates of size at most $k = \Theta(d)$ and such that:*

*(i) it first makes $n^{O(k)}$ queries to MAX-SAMPLE oracle (or MAX-DIST oracle) and then,*

*(ii) for any query $s \in [n]$, $T \subseteq [n] \smallsetminus \{s\}$, it returns, in time $2^{O\left(\frac{|T|\ln(1/\epsilon)}{k}\right)} \cdot \mathrm{poly}(|T|)$ and with probability at least $1 - \delta$, an estimate $\hat{R}_{T \cup \{s\}}(s)$ such that $|\hat{R}_{T \cup \{s\}}(s) - R_{T \cup \{s\}}(s)| \le \epsilon$.*

For $k = \Theta(\sqrt{n\ln(1/\epsilon)})$, Theorem 9 gives an algorithm with a pre-processing time of $n^{O(k)}$ and that can answer any query $(s, T)$ in time $2^{O(|T|\ln(1/\epsilon)/k)}|T|^{O(1)} \le 2^{O(\sqrt{|T|\ln(1/\epsilon)})} \le 2^{O(k)}$. Note that the query-time of this algorithm gets better increasing $k$ (although the pre-processing time gets worse since more slates must be queried).

### 4.3 A simulation algorithm

The pre-processing time of the improper learning algorithm increases with the slate size because the algorithm must be able to reply to *every* query after the pre-processing phase. Suppose, however, that the algorithm knows in advance which slates will be queried; in that case, it can perform the pre-processing phase to satisfy only such requests. In this setting where the queries are known offline (or where the oracles can be called lazily), we can get a faster algorithm.

**Corollary 10** (Simulation algorithm). *Let $R$ be a RUM over $[n]$. Choose any element $s \in [n]$, slate $T \subseteq [n] \smallsetminus \{s\}$, $\epsilon \in (0, 1/3)$, and $\delta \in (0,1)$ such that $\epsilon, \delta \ge \frac{1}{|T|^{O(1)}}$. Then, there exists a randomized algorithm that, for $d \ge \sqrt{|T|\ln(1/\epsilon)}$, accesses slates of size at most $k = \Theta(d)$, makes at most $2^{O\left(\frac{|T|\log\frac{1}{\epsilon}}{k}\right)} \cdot \mathrm{poly}(|T|)$ queries to MAX-DIST oracle (or MAX-SAMPLE oracle), and that with probability at least $1 - \delta$ outputs a value $\hat{R}_{T \cup \{s\}}(s)$ such that $|\hat{R}_{T \cup \{s\}}(s) - R_{T \cup \{s\}}(s)| \le \epsilon$*

Note that by choosing $k = \Theta\left(\frac{|T|}{\ln |T|}\right) \le \Theta\left(\frac{n}{\ln n}\right)$, and *constant* $\epsilon \in (0,1)$, the previous algorithm returns, with high probability and accessing slates of size at most $k$, an approximation to $R_{T \cup \{s\}}(s)$ in polynomial time, for any predetermined $s \in [n], T \subseteq [n] \smallsetminus \{s\}$.

We can also interpret this algorithm in the more general setting of Sherstov [2008]. In such setting, Corollary 10 implies that, for any $n$ events in a probability space, the probability of the intersection of all the events can be well-approximated by a linear combination of polynomially many probabilities of smaller intersections (specifically, each intersection is over at most $O(n/\ln n)$ events).

## 5 Lower bounds

In this section we present a lower bound showing that it is impossible to reconstruct the winning distribution of the full slate by only looking at slates of size $o(\sqrt{n})$, i.e., our reconstruction (Theorem 3) is optimal. For simplicity, we consider the approximation of the winning distribution of the full slate $[n]$, as opposed to any slate $T$ as in Theorem 3.[3]

---

[3]This is w.l.o.g., since we can build our construction for a given slate size $t$, by choosing $n = t$; later, one can add to the same construction as many new items as desired (placing them at the end of the sampled permutation), to get to a total of $N \ge t$ items in the RUM.

Our construction actually shows that, in this $o(\sqrt{n})$-slates setting, it is impossible to approximate the distribution of the full slate even within a constant $\ell_\infty$-error. Specifically, for any constant $\epsilon > 0$, it is not possible to learn whether a special item, $n$, has probability at least $1 - \epsilon$ or at most $\epsilon$ in the full slate by accessing only slates of size $o(\sqrt{n})$.

We will make use of the following result, which is a consequence of the method of dual polynomials (see, e.g., [Bun and Thaler, 2022, Chapter 6]), and was first proved for cryptographic applications in [Bogdanov et al., 2016, Theorem 1]. While the original result considers general Boolean functions, we state it only for the AND function, plugging in the lower bound on the $\epsilon$-approximate degree of the AND proved in [Bun and Thaler, 2015, Proposition 14].

**Theorem 11** (Bogdanov et al. [2016]). *For a sufficiently large $n$ and constant $\epsilon \in (0,1)$, there exists a constant $c > 0$ and two probability distributions $\mu, \psi$ over $\{0,1\}^n$ such that: (i) for each polynomial $p : \{0,1\}^n \to \mathbb{R}$ of degree at most $c \cdot \sqrt{n \cdot \epsilon}$, $\mathrm{E}_{x \sim \mu}[p(x)] = \mathrm{E}_{x \sim \psi}[p(x)]$, and (ii) $|\mathrm{E}_{x \sim \mu}[\mathrm{AND}(x)] - \mathrm{E}_{x \sim \psi}[\mathrm{AND}(x)]| > 1 - \epsilon$*

We are now ready to prove our lower bound for RUMs.

**Theorem 12** (RUMs Lower Bound). *For a sufficiently large $n$ and for constant $\epsilon \in (0,1)$, there exists a constant $c > 0$ and two RUMs $A, B$ on $[n]$ such that: (i) for each $S \subseteq [n]$ such that $|S| \leq c \cdot \sqrt{(n-1) \cdot \epsilon} = \Theta(\sqrt{n})$, it holds $A_S = B_S$, and (ii) $|A_{[n]}(n) - B_{[n]}(n)| > 1 - \epsilon$*

*Proof.* Consider the distributions $\mu, \psi$ over $\{0,1\}^{n-1}$ from Theorem 11, and let $k = c \cdot \sqrt{(n-1) \cdot \epsilon}$. We build RUM $A$ on $[n]$ as follows: sample $x \in \{0,1\}^{n-1}$ according to $\mu$, then, let $S_x = \{i \in [n-1] \mid x_i = 1\}$ and sample a uniform at random permutation among those where the set of elements ranked *lower* than $n$ is $S_x$. RUM $B$ is defined similarly, but $x$ is sampled from $\psi$. Note that for $S \subseteq [n-1]$, $A_{S \cup \{n\}}(n) = \mathrm{E}_{x \sim \mu}[\chi_S(x)]$, and similarly for $B$. Then, from property (ii) of Theorem 11 we have:

$$|A_{[n]}(n) - B_{[n]}(n)| = \left| \mathop{\mathrm{E}}_{x \sim \mu}[\mathrm{AND}_{n-1}(x)] - \mathop{\mathrm{E}}_{x \sim \psi}[\mathrm{AND}_{n-1}(x)] \right| > 1 - \epsilon.$$

Moreover, fix any $S \subseteq [n-1], |S| \leq k$. We have:

$$A_{S \cup \{n\}}(n) = \mathop{\mathrm{E}}_{x \sim \mu}[\chi_S(x)] = \mathop{\mathrm{E}}_{x \sim \psi}[\chi_S(x)] = B_{S \cup \{n\}}(n),$$

where the second equality follows by Theorem 11(ii) and since $\chi_S(x) = \prod_{i \in S} x_i$ is a polynomial of degree $|S| \leq k$.

It remains to show that the winning distributions for the elements different from $n$ in the small slates also coincide. We show that for RUMs $A$ and $B$, these probability distributions can be expressed in terms of winning distributions of $n$; below, we do the calculations only for $A$, the calculations for $B$ are analogous.

Let $\Pi_x$ be the uniform distribution over the set of permutations where the set of elements ranked *lower* than $n$ is $S_x$, for a string $x \in \{0,1\}^{n-1}$. For convenience, for $x \in \{0,1\}^{n-1}$, we set $x_n = 1$. Choose any $i \in S \subseteq [n], i \neq n, |S| \leq k$, then:

$$A_S(i) = \mathop{\mathrm{Pr}}_{\pi \sim A}[\pi(S) = i] = \sum_{T \subseteq S} \mathop{\mathrm{Pr}}_{x \sim \mu, \pi \sim \Pi_x}\left[ \chi_T(x) \prod_{s \in S \setminus T}(1 - x_s) = 1 \cap \pi(S) = i \right]$$

$$= \sum_{T \subseteq S} \mathop{\mathrm{Pr}}_{x \sim \mu}\left[ \chi_T(x) \prod_{s \in S \setminus T}(1 - x_s) = 1 \right] \mathop{\mathrm{Pr}}_{x \sim \mu, \pi \sim \Pi_x}\left[ \pi(S) = i \,\middle|\, \chi_T(x) \prod_{s \in S \setminus T}(1 - x_s) = 1 \right].$$

Here, the third equality follows by the law of total probabilities, partitioning on the possible values of bits $\{x_i\}_{i \in S}$. Since it always holds $x_n = 1$, we assume without loss of generality that either $n \in T$ or $n \notin S$. Given that $\Pi_x$ is uniform, we have:

$$\mathop{\mathrm{Pr}}_{x \sim \mu, \pi \sim \Pi_x}\left[ \pi(S) = i \,\middle|\, \chi_T(x) \prod_{s \in S \setminus T}(1 - x_s) = 1 \right] = \begin{cases} \frac{1}{|T|} & \text{if } i \in T \text{ and } |S \setminus T| = 0 \text{ and } n \notin S \\ 0 & \text{if } i \in T \text{ and } n \in T \\ 0 & \text{if } i \in T \text{ and } |S \setminus T| > 0 \\ \frac{1}{|S \setminus T|} & \text{if } i \in S \setminus T \end{cases}$$

Therefore, thanks to the conditioning, this first probability does not depend on $A$ (the probability is the same if we sample $x$ from $\psi$ and then $\pi$ from $\Pi_x$). Moreover:

$$\Pr_{x \sim \mu}\left[\chi_T(x) \prod_{s \in S \smallsetminus T}(1 - x_s) = 1\right] = \operatorname*{E}_{x \sim \mu}\left[\chi_T(x) \prod_{s \in S \smallsetminus T}(1 - x_s)\right] = \operatorname*{E}_{x \sim \mu}\left[\sum_{P \subseteq S \smallsetminus T}(-1)^{|P|} \cdot \chi_{T \cup P}(x)\right]$$

$$= \sum_{P \subseteq S \smallsetminus T}(-1)^{|P|} \cdot A_{T \cup P \cup \{n\}}(n).$$

Since $A_{T \cup P \cup \{n\}}(n) = B_{T \cup P \cup \{n\}}(n)$, we have $A_S(i) = B_S(i)$ for any $i \in S \subseteq [n]$, $|S| \leq k$.  $\square$

## 5.1 Lower bound when only slates of size $k$ are given

We know that accessing slates of size $2, \ldots, k = \Theta(\sqrt{n})$ is sufficient to approximate the full slate. It is natural to wonder if the slates smaller than $k$ are needed, or if those of size exactly $k$ are enough. A simple observation shows that accessing smaller slates is necessary.

**Observation 13.** *For any $k \leq n$, there exists a RUM $R$ on $[n]$ such that: (i) $R_S(s) = 1/k$ for all $s \in S \in \binom{[n]}{k}$, and (ii) $R_{[n]}(i) = 1/k$ for all $i \in [k]$*

*Proof.* Consider the RUM $R$ that first samples a uniform at random element $i \in [k]$, this element is placed at the top of the permutation, followed by elements $[n] \smallsetminus [k]$ permuted uniformly at random, and finally by the elements $[k] \smallsetminus \{i\}$ permuted uniformly at random. Consider any $s \in S \in \binom{[n]}{k}$. If $s \in [k]$, then $R_S(s) = 1/k$. If instead $s \in [n] \smallsetminus [k]$, let $\alpha = |S \cap [k]| < k$, we have, $R_S(s) = (1 - \alpha/k) \cdot 1/(k - \alpha) = 1/k$. Moreover, for any $i \in [k]$, by construction, $R_{[n]}(i) = 1/k$.  $\square$

Consider $k = \epsilon \cdot n$, for any $\epsilon \in (0, 1)$, and let $R$ be the RUM of the previous construction. Consider now RUM $Q$ that samples a uniform at random permutation over $[n]$. Clearly, $R$ and $Q$ coincide on slates of size $k$, but for all $i \in [k]$, $R_{[n]}(i) = 1/k$ and $Q_{[n]}(i) = 1/n$, and in particular the $\ell_1$-distance between $R_{[n]}$ and $Q_{[n]}$ is $k \cdot (1/k - 1/n) + (n - k)/n = 2 - 2\epsilon$. Therefore, any algorithm accessing only slates of size $k = O(\epsilon \cdot n)$ will incur in an $\ell_1$-error of $1 - \epsilon$ on the full slate. By selecting $k = O(n^c)$, for any $c \in (0, 1)$, we have that any algorithm must incur an $\ell_1$-error of $1 - n^{c-1}$ (and also an $\ell_\infty$-error of $\Omega(\frac{1}{n^c})$) on the full slate. On the other hand, accessing all the slates of size $O\left(\sqrt{n \log n}\right)$ and smaller, one can obtain an $\ell_1$-error as small as $1/n^d$ for any constant $d > 0$. Therefore, accessing smaller slates is necessary.

## 6 Fractional $k$-deck

The $k$-deck problem [Kalashnik, 1973] is a well-studied problem at the intersection of combinatorics and computer science. Given a string $s \in \{0, 1\}^n$ and a set $I = \{i_1, \ldots, i_k\} \in \binom{[n]}{k}$, with $i_1 < \cdots < i_k$, the projection of $s$ to $I$, denoted as $s_I$, is the string $s_I = s_{i_1} \ldots s_{i_k}$. The $k$-*deck* of $s$ is the multiset

$$D_k(s) = \left\{s_I \mid \forall I \in \binom{[n]}{k}\right\}.$$

The $k$-deck problem asks for the smallest $k = k(n)$ such that any $n$-bit string can be reconstructed from its $k$-deck. This problem has a long history. Manvel et al. [1991] originally showed that reconstruction is possible with $k = \frac{n}{2}$; they also showed that it is not possible with $k = \Theta(\log n)$. Later, it was shown in [Krasikov and Roditty, 1997] (see also [Scott, 1997]) that reconstruction is always possible with $k = O(\sqrt{n})$; it is widely conjectured that this bound is tight. The best known lower bound, however, is no better than subpolynomial: in [Dudík and Schulman, 2003] it is shown that $k = e^{\Theta(\sqrt{\log n})}$ is insufficient for reconstruction. In [Chierichetti et al., 2018a], lower bounds for the $k$-deck problem were used to obtain lower bounds on the maximum size of slates required for reconstructing a RUM.

In this section we define a fractional version of the $k$-deck problem and show a reconstruction lower bound. Note that the $k$-deck of a string $s$ is a function $f_s : \{0, 1\}^k \to \mathbb{Z}^{\geq 0}$, where $f_s(s')$ is the multiplicity of $s'$ in $D_k(s)$. Now, given a probability distribution $P$ over $n$-bit strings, the

*fractional k-deck* of $P$ is a function $f_P : \{0, 1\}^k \to \mathbb{R}^{\geq 0}$, where $f_P(s') = \mathrm{E}_{s \sim P}[f_s(s')]$. Also, for a distribution $P$ over $n$-bit strings, we define its *ith marginal* to be the probability that the $i$th bit of a string sampled from $P$ equals 1. The fractional $k$-deck problem seeks the minimum $k = k(n, \epsilon)$ such that the fractional $k$-deck of an unknown distribution $P$ over $n$-bit strings is sufficient to approximate any marginal of $P$ to within an additive $\epsilon$. It is easy to see that the fractional $k$-deck problem generalizes the $k$-deck problem (by also setting $\epsilon < 1/2$).

Based on our RUM lower bound, we can construct two probability distributions over binary strings giving the same fractional $\Theta(\sqrt{n})$-deck, but very different marginals for the first bit, obtaining:

**Theorem 14.** *For sufficiently large $n$, and for each constant $\epsilon > 0$, there exists a constant $c > 0$ and two distributions $X_A$ and $X_B$ over $n$-bits strings of weight 1 that (i) give rise to the same fractional $k$-deck, for $k = \lfloor c \cdot \sqrt{n} \rfloor$, (ii) the marginal of the first bit of $X_A$ is $\geq 1 - \epsilon$, and (iii) the marginal of the first bit of $X_B$ is $\leq \epsilon$.*

## 7 Fractional trace reconstruction

In the *trace reconstruction problem*, there is an unknown $n$-bit string $x$ and a parameter $d \in (0, 1)$. A sample is obtained by passing $x$ through a $d$-deletion channel that erases each bit of $x$ independently with probability $d$. In this setting, one asks for the minimum number of samples necessary to reconstruct $x$. Indeed, with probability $(1 - d)^n$, $x$ itself is returned as a sample and hence reconstruction is trivial with $\Omega\left((1 - d)^{-n}\right)$ samples. This problem has been the subject of intense study [Chase, 2021a,b, De et al., 2017, Batu et al., 2004, Holenstein et al., 2008, Nazarov and Peres, 2017, Peres and Zhai, 2017, Viswanathan and Swaminathan, 2008]. It can be solved with $e^{O\left(\sqrt[3]{\frac{n}{1-d}}\right)}$ samples in the so-called high deletion rate setting ($\frac{1}{2} \leq d \leq 1 - \Omega\left(\frac{1}{\sqrt{n}}\right)$), and this bound is tight for a special class of "mean-based" algorithms [De et al., 2017].

In this section we prove an unconditional lower bound for the fractional trace reconstruction problem, which we define similarly to the fractional $k$-deck problem: given a distribution over the $n$-bit strings, sample a string from that distribution, pass the string through a $d$-deletion channel, and return the resulting subsequence as a sample. Note that, each time, a new fresh string is sampled before passing it through the deletion channel. The goal is to reconstruct the marginals of the distribution.

Several variants of trace reconstruction have been studied in the literature [Chen et al., 2023, Davies et al., 2021]. The variant closest to ours is perhaps the average-case trace reconstruction [Peres and Zhai, 2017]. However, average-case trace reconstruction is a computationally easier version of the problem since the single hidden string is sampled from a uniform distribution. Our fractional trace reconstruction is instead a true generalization of trace reconstruction.

We will show that fractional trace reconstruction cannot be solved with fewer than $e^{o(\sqrt{n})}$ samples. Our lower bound is obtained as a corollary of our result for the fractional $k$-deck problem.

**Theorem 15.** *For each constant $\epsilon > 0$, there exists a constant $c > 0$ and two distributions $X_A$ and $X_B$ over $n$-bits strings of weight 1 such that if $d = 1 - \frac{c}{2\sqrt{n}} + \frac{1}{2n}$, then (i) with fewer than $e^{o((1-d) \cdot n)} = e^{o(\sqrt{n})}$ samples, the probability of correctly distinguishing between $X_A$ and $X_B$ is at most $\frac{1}{2} + o(1)$, (ii) the marginal of the first bit of $X_A$ is $\geq 1 - \epsilon$, and (iii) the marginal of the first bit of $X_B$ is $< \epsilon$.*

In this very high deletion rate setting ($d = 1 - \Theta\left(\frac{1}{\sqrt{n}}\right)$), the magnitude of our lower bound for fractional trace reconstruction is not larger than that of [De et al., 2017] for trace reconstruction; our lower bound, though, holds for any reconstruction algorithm, not just for mean-based ones.

## 8 Conclusions

We considered the problem of learning a RUM by only looking at slates of size at most $k$. We showed that to obtain a uniform approximation of the winning distributions, $k = \Theta(\sqrt{n})$ is necessary and sufficient. Moreover, we provided two explicit algorithms that learn the RUM. While optimal with respect to the slate-size, both our algorithms require time exponential in $n$: we leave open the problem of finding algorithms with better running times.

We also provided a third algorithm that can approximate any given slate by accessing only polynomially many subslates of size at most $\Theta(n/\ln n)$. In this setting, we leave open the problem of decreasing the slate size, while maintaining a polynomial running time.

Another interesting research direction would be considering a PAC-learning variant of the problem, where the slates of the testing phase and/or the training phase come from a probability distribution and the goal is to minimize the expected error in the testing phase. This variant might be easier with respect to both slate size and computational complexity.

## Acknowledgments and Disclosure of Funding

The authors thank the anonymous reviewers, whose insightful suggestions directly led to a significant simplification of our results. Flavio Chierichetti and Alessandro Panconesi were supported in part by BiCi – Bertinoro international Center for informatics and by a Google Focused Research Award. Flavio Chierichetti was supported in part by the PRIN project 20229BCXNW.

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

# A Missing proofs

## A.1 Proof of improper learning algorithm result

*Proof of Theorem 9.* Fixed integer $d \geq \sqrt{n \ln(1/\epsilon)}$, consider the polynomials given by Theorem 8 to approximate $\mathrm{AND}_d, \ldots, \mathrm{AND}_n$. These polynomials will have degrees $k_d, \ldots, k_n$ and $\ell_1$-norm of the coefficients $W_d, \ldots, W_n$. Let $k - 1 = \max\{k_i\} = \Theta(d)$ be the maximum degree of these polynomials and let $W = \max\{W_i\} \leq 2^{O(\frac{n \ln(1/\epsilon)}{k})}$ be the maximum $\ell_1$-norm of the coefficients of these polynomials. Note that these values can be computed in $\mathrm{poly}(n)$ time. The algorithm will access slates of size at most $k$.

Let us start by describing the first phase. Using the MAX-DIST oracle, the algorithm simply stores the values $\{R_S(s)\}_{s \in S \subseteq [n], |S| \leq k}$. Using the MAX-SAMPLE oracle, the algorithm must instead estimate such values. In particular, let $H = k \cdot \sum_{c=0}^{k} \binom{n}{c}$, $\alpha = \lceil \max\{1, \log_{\sqrt{n}} \frac{1}{\epsilon}, \log_{\sqrt{n}} \frac{2}{\delta}\} \rceil$, and $m = \lceil W \rceil^4 \cdot n^{4\alpha} + H$, note that $\alpha = O(1)$ given that $\epsilon\delta \geq 1/n^{O(1)}$, therefore $m = n^{O(k)}$. Fixed any $s \in [n], S \subseteq [n] \setminus \{s\}, |S| \leq k - 1$, the algorithm makes $m$ queries to MAX-SAMPLE oracle for the slate $S \cup \{s\}$. Upon such query the oracle samples $\pi \sim R$ and returns $\pi(S \cup \{s\})$, let $X_i$ be the indicator variable that is 1 if the $i$th query returns $s$ and 0 otherwise. Define the random variable $\overline{R}_{S \cup \{s\}}(s) = \frac{1}{m} \sum_{i \in [m]} X_i$, and observe that $\mathrm{E}\left[\overline{R}_{S \cup \{s\}}(s)\right] = R_{S \cup \{s\}}(s)$. A standard additive Chernoff bound (see, e.g., [Dubhashi and Panconesi, 2009, Theorem 1.1]) ensures that:

$$\Pr\left[|\overline{R}_{S \cup \{s\}}(s) - R_{S \cup \{s\}}(s)| \geq \sqrt{\frac{\ln m}{m}}\right] \leq \frac{2}{m^2}.$$

We repeat this process for each slate of size at most $k$ and, given that $m \geq H$, a simple union bound ensures that with probability at least $1 - 2/m \geq 1 - \delta$ it holds that, for each $s \in [n], S \subseteq [n] \setminus \{s\}, |S| \leq k - 1$, we have $|\overline{R}_{S \cup \{s\}}(s) - R_{S \cup \{s\}}(s)| \leq \sqrt{\frac{\ln m}{m}} \leq m^{-1/4}$. This concludes the first phase of the algorithm, which makes $n^{O(k)}$ queries and is successful with probability $1 - \delta$.

Consider now a query $s \in [n], T \subseteq [n] \setminus \{s\}$, we assume $|T| \geq k \geq \sqrt{n}$ otherwise the algorithm can just return $\overline{R}_{T \cup \{s\}}(s)$ (or $R_{T \cup \{s\}}(s)$ if it has access to MAX-DIST oracle). We describe the rest of the algorithm with the MAX-SAMPLE oracle. Consider the polynomial, given by Theorem 8, of degree at most $k - 1$ with coefficients $\{a_i\}_{0 \leq i \leq k-1}$ that $\epsilon$-approximates $\mathrm{AND}_{|T|}$ and let $\overline{W} = \sum_{c=0}^{k-1} \binom{|T|}{c} |a_c| \leq W$. Note that the inequalities are true by our initial choice of $k$ and $W$. Define:[4]

$$\overline{R}_{T \cup \{s\}}(s) = \sum_{S \subseteq T, |S| \leq k-1} a_{|S|} \cdot \overline{R}_{S \cup \{s\}}(s).$$

By the triangle inequality and Observation 7, we have:

$$|\overline{R}_{T \cup \{s\}}(s) - R_{T \cup \{s\}}(s)| \leq |\overline{R}_{T \cup \{s\}}(s) - \widetilde{R}_{T \cup \{s\}}(s)| + |\widetilde{R}_{T \cup \{s\}}(s) - R_{T \cup \{s\}}(s)|$$
$$\leq \sum_{S \subseteq T, |S| \leq k-1} |a_{|S|}| \cdot |\overline{R}_{S \cup \{s\}}(s) - R_{S \cup \{s\}}(s)| + \epsilon$$
$$\leq \overline{W} \cdot m^{-1/4} + \epsilon \leq 2\epsilon.$$

Grouping terms with the same coefficient, we have:

$$\overline{R}_{T \cup \{s\}}(s) = \sum_{c=0}^{k-1} \left( a_c \cdot \sum_{S \in \binom{T}{c}} \overline{R}_{S \cup \{s\}}(s) \right) = \sum_{c=0}^{k-1} a_c \binom{|T|}{c} \underset{S \sim \binom{T}{c}}{\mathrm{E}} \left[ \overline{R}_{S \cup \{s\}}(s) \right],$$

where the expectation is over a uniform at random $S \in \binom{T}{c}$. To get a faster algorithm, we estimate this expectation via sampling. Fix $c \in \{0, 1, \ldots, k-1\}$, and define $\overline{m} = \lceil \overline{W}^4 \rceil \cdot |T|^{4\alpha}$. Sample

---

[4]When using MAX-DIST oracle we simply have $\overline{R}_{T \cup \{s\}}(s) = \widetilde{R}_{T \cup \{s\}}(s)$

$\overline{m}$ slates $S_1, \ldots, S_{\overline{m}}$ i.i.d. and uniformly at random from $\binom{T}{c}$. Define $M_c = \frac{1}{\overline{m}} \sum_{i=1}^{\overline{m}} \overline{R}_{S_i \cup \{s\}}(s)$. Again via a Chernoff–Hoeffding bound, we obtain:

$$\Pr\left[ |M_c - \mathop{\mathrm{E}}_{S \sim \binom{T}{c}} \left[ \overline{R}_{S \cup \{s\}}(s) \right]| \geq \sqrt{\frac{\ln \overline{m}}{\overline{m}}} \right] \leq \frac{2}{\overline{m}^2}.$$

Using that $\overline{m} \geq |T| \geq k$ and by applying a union bound, we obtain that with probability at least $1 - 2/\overline{m} \geq 1 - \delta$, for each $c \in \{0, 1, \ldots, k-1\}$ it holds $|M_c - \mathrm{E}_{S \sim \binom{T}{c}} \left[ \overline{R}_{S \cup \{s\}}(s) \right]| \leq \sqrt{\frac{\ln \overline{m}}{\overline{m}}} \leq \overline{m}^{-1/4} \leq \overline{W}^{-1} |T|^{-\alpha}$. Define finally our estimate for $R_{T \cup \{s\}}(s)$:

$$\hat{R}_{T \cup \{s\}}(s) = \sum_{c=0}^{k-1} a_c \binom{|T|}{c} \cdot M_c.$$

With probability at least $1 - \delta$, we have:

$$\left| \hat{R}_{T \cup \{s\}}(s) - \overline{R}_{T \cup \{s\}}(s) \right| \leq \sum_{c=0}^{k-1} |a_c| \binom{|T|}{c} \left| M_c - \mathop{\mathrm{E}}_{S \in \binom{T}{c}} \left[ \overline{R}_{S \cup \{s\}}(s) \right] \right| \leq \overline{W} \cdot \overline{W}^{-1} \cdot |T|^{-\alpha} \leq \epsilon.$$

Therefore, a triangle inequality implies $|\hat{R}_{T \cup \{s\}}(s) - R_{T \cup \{s\}}(s)| \leq 3\epsilon$. Since $\epsilon$ can be chosen arbitrarily, we can set $\epsilon := \epsilon/3$ and obtain the desired approximation guarantee.

Finally, note that by Theorem 8, $\overline{W} \leq 2^{O\left( \frac{|T| \ln(1/\epsilon)}{k} \right)}$, therefore $k \cdot \overline{m} = 2^{O\left( \frac{|T| \ln(1/\epsilon)}{k} \right)} \cdot \mathrm{poly}(|T|)$, and this is also the running time to answer a query. □

## A.2 Proof of simulation algorithm result

*Proof of Corollary 10.* Consider the polynomial given by Theorem 8 to approximate $\mathrm{AND}_{|T|}$. Such polynomial has degree $k = \Theta(d)$, and $\ell_1$-norm of the coefficients at most $W = 2^{O\left( \frac{|T| \ln(1/\epsilon)}{k} \right)}$. Setting $\overline{m} = \lceil W^4 \rceil \cdot |T|^{4\alpha}$, for $\alpha = \lceil \max\{1, \log_{|T|} \frac{1}{\epsilon}, \log_{|T|} \frac{2}{\delta}\} \rceil$, we have that the estimate $\hat{R}_{T \cup \{s\}}(s)$ returned by Theorem 9 approximates $R_{T \cup \{s\}}(s)$. To compute such an estimate, one needs to compute $M_c$ for each $c \in \{0, 1, \ldots, k-1\}$, and each of these values requires knowing $\overline{R}_{S_i \cup \{s\}}(s)$ for $\overline{m}$ slates $S_1, \ldots, S_{\overline{m}}$, finally, each such slate requires $m = k \cdot \overline{m}$ queries to MAX-SAMPLE oracle (or one query to MAX-DIST oracle). Therefore, with a total of $k \cdot \overline{m} \cdot m = 2^{O\left( \frac{|T| \ln(1/\epsilon)}{k} \right)} \cdot \mathrm{poly}(|T|)$ queries, we can provide the desired approximation. □

## A.3 Proof of fractional $k$-deck lower bound

As a first step to proving a lower bound for the fractional $k$-deck problem, we show that, for the RUMs of our lower bound construction, the MAX-DIST oracle can be used to reconstruct, for each slate, the distribution over the permutations of the slate induced by the RUM.[5]

**Lemma 16.** *Consider RUMs $A$ and $B$ over $[n]$ given by Theorem 12. For each slate $S$, the distribution over the permutations of $S$ induced by RUM $A$ (resp. $B$) can be obtained from the values $\{A_{T \cup \{n\}}(n)\}_{T \subseteq S}$ (resp. $\{B_{T \cup \{n\}}(n)\}_{T \subseteq S}$).*

*Proof.* Fix any slate $S \subseteq [n]$. For a permutation $\pi$ over $[n]$, denote with $S_\pi$ the slate $S$ sorted according to $\pi$. Let $\mathcal{S}$ be the set of all possible permutations of $S$. Note that $S_\pi \in \mathcal{S}$. We focus on RUM $A$, the proof is identical for $B$. We have to prove that for all $p \in \mathcal{S}$, $\Pr_{\pi \sim A}[S_\pi = p]$ can be written in terms of $\{A_{T \cup \{n\}}(n)\}_{T \subseteq S}$. It holds:

$$\Pr_{\pi \sim A}[S_\pi = p] = \sum_{T \subseteq S} \mathop{\mathrm{Pr}}_{x \sim \mu, \pi \sim \Pi_x} \left[ \chi_T(x) \prod_{s \in S \setminus T} (1 - x_s) = 1 \cap S_\pi = p \right]$$

$$= \sum_{T \subseteq S} \mathop{\mathrm{Pr}}_{x \sim \mu} \left[ \chi_T(x) \prod_{s \in S \setminus T} (1 - x_s) = 1 \right] \mathop{\mathrm{Pr}}_{x \sim \mu, \pi \sim \Pi_x} \left[ S_\pi = p \,\middle|\, \chi_T(x) \prod_{s \in S \setminus T} (1 - x_s) = 1 \right].$$

[5]Incidentally, this is impossible in general RUMs.

As proved in Theorem 12, the first probability can be written in terms of $\{A_{P\cup\{n\}}(n)\}_{P\subseteq S}$. It is also easy to see that, once the bits $\{x_i\}_{i\in S}$ are fixed, one can write explicitly the probability that $S_\pi = p$, and that such probability does not depend on RUM $A$. $\qquad\square$

We can now prove the lower bound for fractional $k$-deck.

*Proof of Theorem 14.* We use the RUMs $A$ and $B$ from the construction of Theorem 12 to obtain two distributions over $n$-bit strings of Hamming weight one.

Given RUM $A$ (resp., $B$) we define $X_A$ (resp., $X_B$) to return an $n$-bit string containing a 1 in position $i$, and 0's in all the other positions, whenever $A$ (resp., $B$) returns a permutation where the "special" item $n$ is in position $i$.

Recall that if $0 < \epsilon < \frac{1}{2}$ is a constant, Theorem 12 guarantees that for each slate $S \subseteq [n]$ such that $|S| \leq c_\epsilon \cdot \sqrt{n}$ for some $c_\epsilon > 0$, it holds $A_S = B_S$. As a consequence, Lemma 16 guarantees that RUMs $A$ and $B$ give rise to the same distributions over projected permutations on slates of size at most $k = \lfloor c_\epsilon \cdot \sqrt{n} \rfloor$. Thus, the fractional $k$-deck of $X_A$ equals the fractional $k$-deck of $X_B$. Indeed, let $S_\pi$ be the binary string induced by slate $S$ when sorted according to $\pi$, then, for any $s \in \{0,1\}^k$,

$$f_{X_A}(s) = \sum_{S\in\binom{[n]}{k}} \Pr_{\pi\sim A}[S_\pi = s] = \sum_{S\in\binom{[n]}{k}} \Pr_{\pi\sim B}[S_\pi = s] = f_{X_B}(s).$$

Theorem 12 also entails that $A_{[n]}(n) \geq 1 - \epsilon$ and $B_{[n]}(n) \leq \epsilon$. Therefore, $X_A$ returns a bit string with a 1 in the first position with probability at least $1 - \epsilon$, whereas $X_B$ returns a bit string with a 1 in the first position with probability at most $\epsilon$. This yields the desired lower bound. $\qquad\square$

### A.4   Proof of fractional trace reconstruction lower bound

*Proof of Theorem 15.* Given a string $x \in \{0,1\}^n$, let $D_{d,x}$ be the random variable that represents the output of a deletion channel with erasure probability $d$ on input $x$; let $\xi_{\alpha,d,x}$ be the event that this output has at least $\alpha(1-d)n$ bits. Then, $\Pr[\xi_{\alpha,d,x}]$ is a function only of $\alpha, d$, and $|x| = n$; let $p_{\alpha,d,n} = \Pr[\xi_{\alpha,d,x}]$.

**Observation 17.** *It holds $p_{2,d,n} \leq e^{-(1-d)\cdot n/3}$.*

*Proof.* Let $x$ be any string of length $n$. By a standard multiplicative Chernoff bound (see, e.g., [Dubhashi and Panconesi, 2009, Theorem 1.1]), we get:

$$\Pr\left[|D_{d,x}| \geq 2(1-d)n\right] = \Pr\left[|D_{d,x}| \geq 2\,\mathrm{E}\left[|D_{d,x}|\right]\right] \leq e^{-\mathrm{E}[|D_{d,x}|]/3} = e^{-(1-d)n/3}. \qquad\square$$

Thus, if we sample the deletion channel fewer than $o\left(e^{(1-d)\cdot n/3}\right)$ times, with probability $1 - o(1)$ we will never get a sample of length more than $2(1-d)n$. We can then apply Theorem 14, which gave the same fractional $k$-deck (hence the same fractional $k'$-deck,[6] for each $1 \leq k' \leq k$) for $k = \lfloor c \cdot \sqrt{n} \rfloor$ to its two distributions $X_A$ and $X_B$, while guaranteeing very different marginals on their first bit. Selecting $d = 1 - \frac{c}{2\sqrt{n}} + \frac{1}{2n}$, we get that $2(1-d)n = c \cdot \sqrt{n} + O(1)$ and therefore we have no sample of length more than $c \cdot \sqrt{n} + O(1)$. Under this conditioning, the two distributions induced by the channel seeded by $X_A$ and the channel seeded by $X_B$ are the same. Indeed, call $\Psi$ such event, then for any string $y$, $|y| \leq 2(1-d)n$,

$$\Pr_{X\sim X_A}[D_{d,X} = y \mid \Psi] = \Pr_{X\sim X_A}[|D_{d,X}| = |y| \mid \Psi] \cdot \Pr_{X\sim X_A}[D_{d,X} = y \mid |D_{d,X}| = |y|]$$

$$= \Pr_{X\sim X_B}[|D_{d,X}| = |y| \mid \Psi] \cdot \sum_{x\in\{0,1\}^n} \Pr_{X\sim X_A}[X = x] \cdot \frac{f_x(y)}{\binom{n}{|y|}}$$

$$= \frac{\Pr_{X\sim X_B}[|D_{d,X}| = |y| \mid \Psi]}{\binom{n}{|y|}} \cdot f_{X_A}(y)$$

---

[6]This is true for standard $k$-deck (see [Manvel et al., 1991]) and the argument directly generalizes to the fractional version.

$$= \frac{\Pr_{X \sim X_B}[|D_{d,X}| = |y| \mid \Psi]}{\binom{n}{|y|}} \cdot f_{X_B}(y)$$

$$= \Pr_{X \sim X_B}[D_{d,X} = y \mid \Psi].$$

We then get the desired lower bound. $\qquad\square$

## B  Proper learning algorithm with MAX-SAMPLE oracle

**Theorem 18.** *Let $P$ and $Q$ be two RUMs on $[n]$. There exists constants $c, c' > 0$ such that for a given $s \in [n]$, $T \subseteq [n] \smallsetminus \{s\}$, if $|P_{S \cup \{s\}}(s) - Q_{S \cup \{s\}}(s)| \leq 2^{-c' \cdot \sqrt{|T| \ln(1/\epsilon)}}$ for each $S \in \left\{ T' \mid T' \subseteq T \text{ and } |T'| \leq c \cdot \sqrt{|T| \cdot \ln \frac{1}{\epsilon}} \right\}$, then $|P_{T \cup \{s\}}(s) - Q_{T \cup \{s\}}(s)| \leq \epsilon$, where $\epsilon \in (2^{-\sqrt{|T|}+2}, 1/3)$.*

*Proof.* Consider the polynomial given by Theorem 8 for $d = \sqrt{|T| \ln(1/\epsilon)}$, of degree $k \leq c \cdot \sqrt{|T| \ln(1/\epsilon)} = \Theta(\sqrt{|T| \ln(1/\epsilon)})$, and with $\ell_1$-norm of the coefficients $W \leq 2^{c'' \cdot \sqrt{|T| \ln(1/\epsilon)}} = 2^{O(\sqrt{|T| \ln(1/\epsilon)})}$ for some constants $c, c'' > 0$, that $\epsilon$-approximates $\text{AND}_{|T|}$. Set $c' = c'' + 1$. Then, by Observation 7,

$$\begin{aligned}
\left| P_{T \cup \{s\}}(s) - Q_{T \cup \{s\}}(s) \right| \leq & |P_{T \cup \{s\}}(s) - \widetilde{P}_{T \cup \{s\}}(s)| + |\widetilde{Q}_{T \cup \{s\}}(s) - Q_{T \cup \{s\}}(s)| + \\
& |\widetilde{P}_{T \cup \{s\}}(s) - \widetilde{Q}_{T \cup \{s\}}(s)| \\
\leq & 3\epsilon,
\end{aligned}$$

where the last inequality uses $|\widetilde{P}_{T \cup \{s\}}(s) - \widetilde{Q}_{T \cup \{s\}}(s)| \leq W \cdot 2^{-c' \sqrt{|T| \ln(1/\epsilon)}} \leq 2^{-\sqrt{|T|}} \leq \epsilon$. Setting $\epsilon := \epsilon/3$ completes the proof. $\qquad\square$

A practical implication of this theorem is a proper learning algorithm using MAX-SAMPLE oracle. Indeed, using enough samples one can approximate the winning distributions within an exponentially small error, and then, fitting the approximate winning distributions with a RUM minimizing the maximum $\ell_\infty$-error, the theorem guarantees that such RUM will approximate the original RUM on any slate.

It is not difficult to see that an analogous theorem can be proved in the more general setting of Sherstov [2008]: taken any events $A_1, \ldots, A_n$, and $B_1, \ldots, B_n$ such that the probabilities of the intersections up to size $\Theta(\sqrt{n \ln(1/\epsilon)})$ differ by at most an exponentially small error, we have that the probabilities of any intersection differ by at most $\epsilon$.

## C  Proper learning algorithm in time $2^{O(n)}$

In this section, we give more details on how to improve the running time of the proper learning algorithm from $n^{O(n)}$ to $2^{O(n)}$, both for the MAX-DIST oracle and the MAX-SAMPLE oracle. The $n^{O(n)}$ algorithm for both oracles, can be obtained by solving directly the following LP with the ellipsoid method, for $k = \Theta(\sqrt{n \ln(1/\epsilon)})$.

$$\begin{cases}
\min \epsilon & \\
\sum_{\substack{\pi \in \mathbf{S}_n \\ \pi(S \cup \{s\}) = s}} p_\pi - D(S \cup \{s\}, s) \leq \epsilon & \forall s \in [n] \; \forall S \subseteq [n] \smallsetminus \{s\} \text{ s.t. } |S| \leq k - 1 \\
\sum_{\substack{\pi \in \mathbf{S}_n \\ \pi(S \cup \{s\}) = s}} p_\pi - D(S \cup \{s\}, s) \geq -\epsilon & \forall s \in [n] \; \forall S \subseteq [n] \smallsetminus \{s\} \text{ s.t. } |S| \leq k - 1 \\
\sum_{\pi \in \mathbf{S}_n} p_\pi = 1 & \\
\epsilon \geq 0 & \\
p_\pi \geq 0 & \forall \pi \in \mathbf{S}_n
\end{cases} \tag{2}$$

Where $D(S \cup \{s\}, s)$ is equal to $Q_{S \cup \{s\}}(s)$ for a RUM $Q$ in case of MAX-DIST oracle, or it is equal to an approximation $\widetilde{Q}_{S \cup \{s\}}(s)$ in case of MAX-SAMPLE oracle. Our strategy is to compute the optimal solution of (2) by first passing to the dual to reduce the number of variables. In particular, by employing the same ideas of Chierichetti et al. [2023], one can see that the dual LP is:

$$
\begin{cases}
\max B - \displaystyle\sum_{s \in S \subseteq [n], |S| \leq k} D(S, s) \cdot \Delta_{S,s} \\
\displaystyle\sum_{S \subseteq [n], 1 \leq |S| \leq k} \Delta_{S, \pi(S)} \geq B & \forall \pi \in \mathbf{S}_n \\
\displaystyle\sum_{s \in S \subseteq [n], |S| \leq k} |\Delta_{S,s}| \leq 1 \\
\Delta_{S,s} \text{ unrestricted} & \forall s \in S \subseteq [n], |S| \leq k \\
B \text{ unrestricted}
\end{cases}
\tag{3}
$$

Note that the constraint $\displaystyle\sum_{s \in S \subseteq [n], |S| \leq k} |\Delta_{S,s}| \leq 1$ can easily be turned into a linear constraint by introducing one auxiliary variable for each $\Delta_{S,s}$. Given a possible solution $\overline{B}, \{\overline{\Delta}_{S,s}\}_{s \in S \subseteq [n], |S| \leq k}$, checking all the constraints $\{\sum_{S \subseteq [n], 1 \leq |S| \leq k} \overline{\Delta}_{S, \pi(S)} \geq \overline{B}\}_{\pi \in \mathbf{S}_n}$ is equivalent to solving an instance of Weighted Feedback Hyperedge Set as described in Chierichetti et al. [2023] and this can be done in time $2^{O(n)}$ via dynamic programming [Chierichetti et al., 2023, Theorem 10]. Therefore, via the ellipsoid method, the dual (3) can be solved in time $2^{O(n)}$. In particular the ellipsoid method will call the separation oracle at most $2^{O(n)}$ times, obtaining at most $2^{O(n)}$ separating hyperplanes relative to a permutation constraint. Let $P \subseteq \mathbf{S}_n$ be the set of permutations relative to these constraints. Now, by solving the LP (2) restricted to the permutation variables $P$, we obtain an optimal RUM $R$ for LP (2) in time $2^{O(n)}$. The output of our proper learning algorithm is $R$.

## D Computing coefficients in polynomial time

We show in this section that the coefficients of the polynomial proposed by Huang and Viola [2022], to approximate $\text{AND}_n$, are computable in polynomial time. This is folklore in the literature, but for completeness, we provide here an explicit algorithm.

Since the AND is a symmetric function, the construction of Huang and Viola [2022] consists of first building a univariate polynomial $q : [0, 1] \to \mathbb{R}$ such that: (i) $q(1) = 1$, and (ii) $|q(i/n)| \leq \epsilon$ for all $i \in \{0, 1, \ldots, n - 1\}$. Then, the final polynomial is $p(x) = q(\sum_{i=1}^n x_i / n)$ that clearly $\epsilon$-approximates the AND on $n$ bits. We first show that, given the coefficients of the univariate polynomial $q$, those of the final polynomial $p$ are computable in polynomial time.

For a set $A$ and a tuple $T \in A^b$, define $\text{set}(T)$ as the set obtained by removing duplicates from $T$ (e.g., $\text{set}((1, 2, 1, 2, 3)) = \{1, 2, 3\}$). For a set $A \neq \varnothing$ and integer $b \geq 1$, define $m(A, b) = |\{T \in A^b \mid \text{set}(T) = A\}|$. Instead of writing $m([a], b)$ we usually write $m(a, b)$ for integer $a \geq 1$. Define for convenience $m(0, 0) = 1$ and $m(0, b) = 0$ for $b \geq 1$.

**Lemma 19.** *The following properties hold for $A \neq \varnothing, b \geq 1, a \geq 1$:*

*(i) $m(A, b) = m(|A|, b)$,*

*(ii) if $b < a$, $m(a, b) = 0$,*

*(iii) for every $n \geq 0$, $\sum_{i=1}^b \binom{n}{i} m(i, b) = n^b$,*

*(iv) if $b \geq a$, $m(a, b) = a^b - \sum_{i=1}^{a-1} \binom{a}{i} \cdot m(i, b)$.*

*Proof.* Properties (i) and (ii) are immediate by definition.

Note that property (iii) is true for $n = 0$. If $n \geq 1$, note that for each tuple $T \in [n]^b$ there exists a unique non-empty set $S \subseteq [n]$ such that $\text{set}(T) = S$. Let $[\text{set}(T) = S]$ be 1 if $\text{set}(T) = S$ and 0

otherwise. Then:

$$n^b = \sum_{S\subseteq[n],S\neq\varnothing} \sum_{T\in[n]^b} [\mathrm{set}(T)=S] = \sum_{S\subseteq[n],S\neq\varnothing} \sum_{T\in S^b} [\mathrm{set}(T)=S] = \sum_{S\subseteq[n],S\neq\varnothing} m(S,b)$$

$$= \sum_{i=1}^{\min\{n,b\}} \binom{n}{i} m(i,b),$$

where the last equality applies properties (i) and (ii). Property (iv) follows by applying property (iii) with $n=a$. $\qquad\square$

**Lemma 20.** *If $q : [0,1] \to \mathbb{R}$ is a univariate polynomial of degree $k$ with real coefficients $\{\alpha_i\}_{i\in\{0,1,\ldots,k\}}$, then $p : \{0,1\}^n \to \mathbb{R}$, such that $p(x) = q\left(\frac{\sum_{i=1}^n x_i}{n}\right)$, is a multivariate polynomial of degree $k$ such that $p(x) = \sum_{S\subseteq[n]:|S|\leq k} \beta_{|S|} \cdot \chi_S(x)$ with coefficient $\beta_{|S|} = \sum_{b=|S|}^{k} \alpha_b \cdot \frac{m(|S|,b)}{n^b}$.*

*Proof.* By definition, $q\left(\frac{\sum_{i=1}^n x_i}{n}\right) = \sum_{i=0}^{k} \alpha_i \frac{\left(\sum_{j=1}^n x_j\right)^i}{n^i}$. Note that for each $j \in [n]$, $x_j^2 = x_j$, therefore, for $i \geq 1$, we have:

$$\left(\sum_{j\in[n]} x_j\right)^i = \sum_{t\in[n]^i} \prod_{j\in[i]} x_{t_j} = \sum_{S\subseteq[n],1\leq|S|\leq i} m(|S|,i)\chi_S(x),$$

where we used that $m(|S|,i) = m(S,i) = \left|\{t \in [n]^i \mid \mathrm{set}(t) = S\}\right|$. Therefore:

$$p(x) = \alpha_0 + \sum_{i=1}^{k} \sum_{S\subseteq[n],1\leq|S|\leq i} \frac{\alpha_i}{n^i} \cdot m(|S|,i) \cdot \chi_S(x)$$

$$= \alpha_0 + \sum_{S\subseteq[n],1\leq|S|\leq k} \sum_{i=|S|}^{k} \frac{\alpha_i}{n^i} \cdot m(|S|,i) \cdot \chi_S(x) = \alpha_0 + \sum_{S\subseteq[n],1\leq|S|\leq k} \chi_S(x)\beta_{|S|}.$$

Finally, note that using the convention $m(0,0) = 1, m(0,b) = 0$ for $b \geq 1$, we have $\beta_0 = \alpha_0$. $\qquad\square$

From the recurrence of Lemma 19(iv), it is easy to devise a dynamic programming algorithm that computes all the needed values of $m(a,b)$ in time $O(k^3)$, where $k$ is the degree of the univariate polynomial. Therefore, from the coefficients of the univariate polynomial $q$, one can compute those of the final polynomial $p$ in time $O(k^3)$, using Lemma 20.

We now focus on the univariate polynomial of Huang and Viola [2022]. This polynomial is the product of $n$ Chebyshev polynomials shifted and scaled. In particular, recall that the Chebyshev polynomial of the first type of degree $d$, $T_d(x)$, is defined by the following recurrence: $T_0(x) = 1$, $T_1(x) = x$, $T_{d+1}(x) = 2x \cdot T_d(x) - T_{d-1}(x)$. Therefore, via dynamic programming, the coefficients of $T_d(x)$ can be computed in $O(d^2)$. A simple calculation shows that, if $\{\alpha_i\}_{i\in\{0,1,\ldots,d\}}$ are the coefficients of $T_d(x)$, then $T_d(a \cdot x^\gamma + b)$, for integer $\gamma$, is a polynomial of degree $d \cdot \gamma$, where the coefficient of the term $x^{j\cdot\gamma}$ is $\sum_{i=j}^{d} \binom{i}{j} a^j b^{i-j} \alpha_i$ for $j \in \{0,1,\ldots,d\}$, while all other monomials have coefficient 0. Therefore, the coefficients of $T_d(a \cdot x^\gamma + b)$ can be computed in $O(d^2)$ from the $\alpha_i$'s. Let $d_i$, for $i \in [n]$ be the coefficient of the $i$th Chebyshev polynomial in the product of the final polynomial. The final polynomial has degree $k$, therefore computing the coefficients of all the scaled Chebyshev polynomials takes time $\sum_{i=1}^{n} O(d_i^2) \leq \left(\sum_{i=1}^{n} O(d_i)\right)^2 = O(k^2)$. Recall that two polynomials of degree $k$ can be multiplied in time $O(k \log k)$ via the Fast Fourier Transform, then the $n-1$ multiplications can be performed in time $O(nk \log k)$. Putting it all together the coefficients of the polynomial can be computed in time $O(k^2 + nk \log k + k^3) = O(nk \log k + k^3)$.

