# OpenReview forum: "Tight Bounds for Learning RUMs from Small Slates"
_NeurIPS.cc/2024/Conference — NeurIPS 2024 poster_

### Official Review · Reviewer_NbMn · 2024-06-13

**Soundness:** 4
**Presentation:** 4
**Contribution:** 4
**Rating:** 8
**Confidence:** 3

**Summary:**

This paper studies the learning of Random Utility Models (RUMs). A RUM is a probability distribution $P$ over the set of permutations over $[n]$. Fix a permutation $\pi$ and consider any subset $T \subseteq [n]$, known as a *slate*. The winner of the slate (corresponding to $\pi$) is the highest ranked element in $T$ accordint to $\pi$. We can imagine fixing a slate $T$, and consider the distribution of the winner of the slate $T$ as we draw $\pi$ from $P$. In the problem of learning RUMs, one is given a set of slates, together with the corresponding distribution of the winner of that slate. The task is to reconstruct the distribution of the winner of every possible slate.

Specifically, this paper considers the problem when the example slates given to a learning algorithm are at most a size $k$. The main result of the paper is that having access to the winning distributions of all slates of size at most $O(k)$ is necessary and sufficient for the task of learning RUMs. The same result also holds if one has only sample access to the winning distributions of all slates of this size. The authors present two algorithms that achieve the upper bound: 1) a proper algorithm, that constructs a RUM in time $n^{O(n)}$, and thereafter, given any slate, returns an approximation to the winning distribution of that slate in polynomial time 2) an improper algorithm, that does not construct a RUM, but runs in time $n^{O(\sqrt{n})}$, and thereafter, given any slate, returns an approximation to the winning distribution of that slate in time $2^{O(\sqrt{n})}$. The latter algorithm thus, in time $n^{O(\sqrt{n})}$, approximates the winning distribution of the full slate $[n]$ in $l_\infty$, which is an improvement over the previous best running time of $2^{O(n)}$ for the same task, implicit in prior work. The latter algorithm can also be adapted to yield the following guarantee: given a prespecified slate $T \subseteq [n]$, one can approximate the winning distribution of $T$ in $l_\infty$ in time polynomial in $n$, by querying slates of size at most $O(|T|/\log|T|)$. The authors also show that any algorithm that only accesses slates of size $o(\sqrt{n})$ can not successfully learn the winning distribution of the full slate. This shows that their earlier two algorithms are optimal in the maximum size of the slates they access. Finally, the authors define fractional versions of two classic problems in the intersection of combinatorics and computer science: $k$-deck and trace reconstruction. Using their techniques for learning RUMs with small slates, they are able to obtain new results for the fractional versions of both these problems (similar quantitative results would be major breakthroughs for the original problems).

The upper bounds rely crucially on results by Sherstov in 2008, and on the state-of-the-art approximation of the AND function by low degree polynomials due to Huang and Viola, 2022. The lower bound is a reduction to a lower bound on the approximation of the AND function by low-degree polynomials, due to Bogdanov et al. 2016.

**Strengths:**

This paper considers a natural constrained formulation of the problem of learning RUMs on $n$ elements: algorithms that are only given input access to slates of size at most $k$. For such algorithms, what is the smallest $k$ that is both necessary and sufficient? Satisfyingly, the authors give a compete answer to this question: $k=O(\sqrt{n})$ is necessary and sufficient. Given the pre-existing technical machinery by Sherstov, 2008, Huang and Viola, 2022 and Bogdanov et al. 2016 (albeit for problems not directly related to learning RUMs), the connections to RUMs are not too difficult to read and understand, and I found them innovative and cute. I also liked the applications of the the authors' techniques to obtain novel bounds for relaxed versions of classical hard problems in computer science. While I am not closely familiar with the literature on these problems, I imagine the fractional definitions of these problems given by the authors, together with their bounds, might be of interest to people who study these problems. Overall, I find the authors' work to be a strong and compelling study.

**Weaknesses:**

If I am to nitpick: while the technical narration of the problem of learning RUMs given by the authors is clear (I admit that I did not even know this problem before reading the paper, but could understand almost all of the paper), I would have liked to see a little more story-building and motivation around this problem in the introduction. I felt that the authors dived into their technical contributions rather too soon. I would also have liked to see the authors elaborate more on how their work compares to prior work by Chierichetti et al. 2021, 2023. As of now, these works, while seeming to be most relevant, are only cited in passing in the related work. Also, I feel that a few more references could be provided with respect to variations of the $k$-deck and trace reconstruction problems that have been considered in the literature.

**Questions:**

1) The authors do show in Observation 13 that having access to slates of size exactly $k=\Theta(\sqrt{n})$ is also not sufficient for learning the RUM. Might it however be possible that such a result holds for some $k > \Theta(\sqrt{n})$? Say I am given all the slates of size exactly $\Theta(n^{0.6})$?


2) Are similar fractional versions of the $k$-deck or trace reconstruction problems known? I suggest doing a literature survey to cite some works that have considered variants of these problems -- this might make these sections more complete and situate your definitions of the problems better within the literature. For example, here are two works I found with a preliminary google search: 1) Approximate Trace Reconstruction from a Single Trace. Xi Chen, Anindya De, Chin Ho Lee, Rocco A. Servedio, Sandip Sinha, 2022 2) Approximate Trace Reconstruction. Sami Davies, Miklos Z. Racz, Cyrus Rashtchian, Benjamin G. Schiffer, 2020.

**Limitations:**

The authors have satisfactorily addressed the limitations of their work.

---

> ### Author Rebuttal · Authors · 2024-08-05
>
> We thank the reviewer for the suggestions and feedback.  In the revision, we will provide more story-building and motivation around the problem.
>
> Regarding related works on RUMs, in [Chierichetti et al., ICML 2021] they take in input a RUM $R$ and their goal is to output a new RUM $R’$ whose support is only on a linear number of permutations, while maintaining a good approximation of $R$. Therefore, they consider a lossy-compression problem rather than a learning problem, as we do instead.
> In [Chierichetti et al., AISTATS, 2023], they take in input the empirical winning distributions of small slates and their goal is to output a RUM consistent with such input slates. Crucially, they do not require their RUM to work well on slates outside of the input set. Instead, we consider a learning problem where the algorithms must provide good approximations on all the slates, even the large ones not given in input.  We will clarify these better in the revision.
>
> We now address the questions:
>
> 1. Unfortunately, the answer is no.  Having access to all and only the slates of size exactly $O(n^c)$ for any constant $0<c<1$ is not sufficient to reconstruct the winning distribution on the full slate. In particular, Observation 13 entails that access to slates of size $O(\epsilon \cdot n)$ leads to an $\ell_1$-error of at least $1 - \epsilon$ on the full slate. Therefore, selecting $\epsilon = n^{c-1}$ yields the claim. We will mention this in the revision.
>
> 2. Thanks for your suggestion and the references; we will mention some variations of the problems to better situate our work in the literature.
>
> Both papers you mentioned, [Chen et al., 2022] and [Davies et al., 2020], fall into the literature of approximate trace reconstruction. In such variation, one seeks to approximately reconstruct the single hidden string $x$, up to some error in edit distance. In our version we have instead a hidden distribution over strings, rather than a fixed string.
>
> Some results of the first paper you mentioned, [Chen et al., 2022], consider the average-case trace reconstruction. The works in this literature are maybe the most related to our version. However, our problem differs from such variations in two aspects. 1) in average-case trace reconstruction the hidden string is sampled from a uniform distribution, instead, we consider arbitrary distributions, and 2) in average-case trace reconstruction, one first samples a string $x$ and then each trace is obtained by running the same string $x$ through the deletion channel. Instead, in our fractional trace reconstruction, to collect a new trace we first sample a new string $x$ and then run it through the deletion channel. Note that the average-case trace reconstruction is not a generalization of trace-reconstruction, while our fractional trace-reconstruction is a true generalization.
>
> We will add these comments in the revision.

---

> > ### Comment · Reviewer_NbMn · 2024-08-07
> > **Response to rebuttal**
> >
> > Thank you for the response. Please do include the discussion from your response, along with the references, in the updated version. I maintain my score of 8, and believe this work should be accepted. Great work!

---

### Official Review · Reviewer_R6u3 · 2024-07-11

**Soundness:** 4
**Presentation:** 3
**Contribution:** 3
**Rating:** 7
**Confidence:** 4

**Summary:**

This paper studies the problem of learning a Random Utility Model with limited information.  A RUM is a distribution on the symmetric group on $n$ letters and a slate is a nonempty subset of the universe of letters.  The paper studies the problem of learning the RUM given access to the probabilities that a given letter $s$ is most preferred by a sample from the RUM out of all of the elements in a given slate $S$.  The authors apply classical techniques to derive an information theoretic upper bound for the necessary size of a slate in order to learn the RUM to additive error $\epsilon$, which is complemented by a matching lower bound at the end of the paper.  They then provide two algorithms bounding the sample complexity of learning the RUM with such slates: one relying on solving a linear program requiring $n^{O(n)}$ samples and another algorithm relying on polynomial approximations for the Boolean AND.  The latter algorithm is then applied to demonstrate that large slates can be used to learn an RUM in polynomial time to constant accuracy.  The paper concludes with applications of their techniques to two other problems in the intersection of combinatorics and CS theory.

**Strengths:**

The paper lays out a clear and interesting problem and provides a fairly complete solution using classical results in Boolean analysis in novel ways.  The solution is thorough and the presentation is clear and compelling.

**Weaknesses:**

The main weakness in the paper is the exponentially large number of slates required by the algorithm in order to learn the RUM.  The simulation lemma is able to reduce this to polynomial at the cost of increasing the size of the slates, but the time is still exponential in the error.  It would be nice to have a clearer idea of the landscape with respect to if polynomially many slates are sufficient even in a polynomially smaller slate setting.  The authors also heavily use prior technical contributions and it might be useful to better delineate the core technical contributions of the present paper.

**Questions:**

1. Is there some middle ground in the size of the slates, say $\Theta(n^c)$ for $1/2 < c < 1$ that allows for poly algorithms in approximating the RUM?  As it stands there is quite a gap between the exponential algorithms in the case of $\Theta(\sqrt{n})$ and the poly algorithms in the case of $\Theta(n / \log n)$.

2. Are there weaker notions of learning the RUM that allow for either smaller slates or more sample-efficient algorithms?  For example, I could imagine that some of the sample complexity is coming from learning the precise order of very low-preference items in high probability according to the RUM.  If one were to only care about properties of the RUM relevant to some notion of the most preferred items, can the bounds be improved?

3.  It would be nice to rigorously demonstrate that the algorithm in Theorem 5 actually can be made to run with $2^{O(n)}$ slates as opposed to $n^{O(n)}$.  It is poor form to simply state this as a footnote, as future authors may wish to cite this result without being forced to rewrite your entire proof.

**Limitations:**

The limitations have been adequately discussed.

---

> ### Author Rebuttal · Authors · 2024-08-05
>
> We thank the reviewer for the feedback and suggestions.
>
> Exponential time: please see the general comment.
>
> On a high level, the main technical contributions are: (i) relating the approximation degree of the AND function to the slate size to learn a RUM (Theorem 9 and Theorem 12), (ii) relating the $\ell_1$-norm of the coefficients to approximate the AND function to the query time of an algorithm for RUM learning (Theorem 9 and Corollary 10), and (iii) deriving, from the lower bound for RUM learning, lower bounds to fractional versions of other well studied problems (Theorem 14 and Theorem 15). The proofs are non-trivial and the techniques we use come from seemingly unrelated areas such as cryptography and Boolean function approximation. We believe that connecting such distant areas to RUM learning is a contribution by itself. In the revision, we will ensure that our contributions are more clearly highlighted.
>
> We now answer the questions in order:
>
> 1. This is an interesting open problem. We currently do not know the optimal slate size that allows for polynomial time simulation algorithms. (We mention this problem in the Conclusions at line 358.)
>
> 2. This is an excellent point.  Our problem currently asks for a good approximation on *all* possible slates. It is possible that a PAC-learning variant of the problem (where, say, the testing and/or the learning phases have to use slates coming from a distribution) might be easier, both in terms of running time and slate size. We will mention this as a possible future direction.
>
> Regarding the point of focusing only on the most preferred items. If we consider the task of just estimating the winning probability of the most preferred item, then our lower bound (Theorem 12) answers this in the negative: there is one item for which we cannot say if its winning probability is $\ge 1-\epsilon$ or $\le \epsilon$ in the full-slate. Thus, the slate size cannot be improved in this context.
>
> 3. Thank you for the suggestion and apologies for the lack of details; we will add a rigorous proof in the revision. The high level idea is that, instead of solving the primal LP (1) (that takes $n^{O(n)}$ time), we can solve the dual LP. The separation oracle of the dual can be solved in time $2^{O(n)}$ using an algorithm in [Chierichetti et al., 2023], therefore, employing the ellipsoid method provides the desired running time.

---

> > ### Comment · Reviewer_R6u3 · 2024-08-10
> > **Thank you for responding**
> >
> > The authors have done a good job of responding to my questions and I maintain my recommendation to accept.  The theoretical contribution and the new techniques used are of interest.

---

### Official Review · Reviewer_5X2F · 2024-07-12

**Soundness:** 3
**Presentation:** 3
**Contribution:** 3
**Rating:** 5
**Confidence:** 2

**Summary:**

A random utility model is defined by a distribution over permutations of $1,2,\dots,n$. Given a non-empty subset $S \subseteq \{1,2,\dots,n\}$, an oracle stochastically returns which one in $S$ is the highest according to this distribution. This paper considers the problem of estimating a random utility model from winning probabilities for small $S$.
- This paper first proposes an algorithm that queries all $S$ with $|S| = \tilde{O}(\sqrt{n})$ and then estimates a RUM by solving an LP. The authors shows that this estimated model has small error even for large $S$. This algorithm takes $n^{O(n)}$ time.
- Next the authors consider an approach of expressing RUMs by polynomials. The proposed algorithm again queries all $S$ with $|S| = \tilde{O}(\sqrt{n})$. However, as the expression of an estimated RUM does not require $n!$ dimensions, both the estimation and prediction takes $n^{\tilde{O}(\sqrt{n})}$ time.
- The authors also provide a lower bound. That is, if two RUMs coincide on all $S \subseteq [n]$ with $|S| = O(\sqrt{n})$, the error for $S = [n]$ can be very large.
- The last contributions are on the fractional versions of the $k$-deck and trace reconstruction problems. The authors give lower bounds on the sample complexity for these problems based on the RUM lower bound.

**Strengths:**

RUMs are an important decision making model in economics and statistics, and the problem of estimating it from data is significant. For this problem, this paper determines the size of queries (i.e., $|S|$) that is necessary for estimating a RUM by providing both upper and lower bounds. For the upper bound, the main technical contribution is to translate the result on the AND function by Sherstov (2008) to RUMs. For the lower bound, the authors construct a difficult instance by reducing it to the difficult instance on the AND function proposed by Bogdanovet al. (2016). These proofs are decently built on existing results on boolean functions. The lower bounds for fractional $k$-deck and trace reconstruction are also interesting.

**Weaknesses:**

In my opinion, the largest weakness of this paper is the lack of practical insight. The main result claims that estimating a RUM requires $n^{\tilde{O}(\sqrt{n})}$ queries, which is unrealistically large for suggested applications such as web search results. Theoretically, this bound is meaningful; the upper and lower bounds coincide, so this order is tight. However, in practice, I do not know a realistic scenario in which we can query all size $\tilde{O}(\sqrt{n})$ subsets, and the proposed algorithm seems to be designed for the theoretical purpose. Actually, the authors do not provide any experimental result. I think there might be a more appropriate venue other than NeurIPS for this paper. The technical contribution of this paper is mainly to relate RUMs and the AND function, which seems to attract more audience in TCS conferences.

**Questions:**

Is there any possible future direction to develop the results of this paper for more practical applications?

**Limitations:**

The contribution of this paper is only theoretical and does not seem to have a large practical impact for now.

---

> ### Author Rebuttal · Authors · 2024-08-05
>
> We thank the reviewer for the comments. Regarding the computational complexity of the algorithms, please see the general comment. Also note that, while querying all the slates of size $O(\sqrt{n})$ is required to learn the complete RUM, if we are interested only in some target slates, our algorithm becomes more efficient as characterized in Corollary 10.
>
> RUM learning is an important ML problem that is still not well understood theoretically. We believe our work is an attempt to fill this gap and would be of interest to the ML community, as evidenced by previous work [Farias et al., NeurIPS 2009, Soufiani et al., NeurIPS 2012, Oh and Shah, NeurIPS 2014, Chierichetti et al., ICML 2018, ICML 2021, Almanza et al., ICML 2022].

---

> > ### Comment · Reviewer_5X2F · 2024-08-09
> >
> > Thank you for the feedback. Although I understand that the improvement made by this paper accelerates the running time so much, but I am still not convinced that the running time $n^{\tilde{O}(\sqrt{n})}$ is practical. Those papers about RUMs published in ICML and NeurIPS mainly considered a more restricted model expressing RUMs and proposed practical algorithms, which is a more promising direction for developing practical algorithms in my opinion. I keep the current borderline accept score for evaluating this paper's theoretical contribution.

---

### Official Review · Reviewer_19Lt · 2024-07-14

**Soundness:** 2
**Presentation:** 2
**Contribution:** 2
**Rating:** 4
**Confidence:** 2

**Summary:**

The paper considers the Random Utility Model (RUM) problem and gives upper and lower bound on plate size.

RUM is a classic economic model that is used to understand user behavior by modeling choices from subsets of available items. In the RUM problem, there is a set of element $[n]$ and there is a probability distribution $P$ over the permutation of $[n]$. The algorithm could query a set of $k$ elements, and either observe the one with the highest rank (sampled from $P$) or the winning distribution on these $k$ elements. The paper seeks to understand the optimal value of $k$, such that the algorithm could figure out the permutation distribution (up to small error) given all plate of size $k$. The answer turns out to be $k = \Theta(\sqrt{n})$.

In particular,
1.  The upper bound shows that knowledge of choices from slates of size$ O(\sqrt{n})$ allows approximating the choice distribution for any slate with high accuracy. Moreover, it gives an algorithm that learns RUMs efficiently: A proper learning algorithm with $n^{n}$ time and an improper learning algorithm with $n^{\sqrt{n}}$ time.
2.  Lower bounds are derived, indicating that learning from slates smaller than $\sqrt{n}$ results in high prediction errors. These results also contribute to understanding the k-deck and trace reconstruction problems.

In terms of technique, the proofs rely on connections between RUM learning and the approximation of Boolean functions by polynomials. The paper leverages results from the approximation of the AND function to develop their bounds and algorithms.

Overall, the paper solves a meaningful problem and characterizes the optimal plater size. The drawback is that that query complexity and the runtime is exponential.

**Strengths:**

The paper solves a meaningful problem and characterize the optimal plater size.

**Weaknesses:**

1. Query complexity and the runtime is exponential;
2. I think it needs some explanation on the query model. In particular, why the model can only see plate of size $k$.
3. The writing could be improved: for example, Line 58, our results are proved *by* observing; Line 95, the citation appears at a wrong place.

**Questions:**

.

**Limitations:**

.

---

> ### Author Rebuttal · Authors · 2024-08-05
>
> We thank the reviewer for the comments.
>
> 1. Please see the general comment.
>
> 2. We were motivated to consider this model by the observation that modern user interfaces work hard to present small and easily scanned slates for the user to consider; typical examples of such interactions are the 10 blue links of Web search, the 3 choices in Google local search, or the handful of movies in a Netflix carousel. Therefore, we allow access only to slates of size $2,\dots, k$ and our goal is to make $k$ as small as possible while providing good estimates to every slate.
>
> 3. Thank you for pointing out the typos; we will improve the writing in the revision.

---

> > ### Comment · Reviewer_19Lt · 2024-08-12
> >
> > Thanks for your reply. I acknowledge that i have read the rebuttal. I will read other reviewers' comments and have discussion with them.

---

### Author Rebuttal · Authors · 2024-08-05

We thank the reviewers for their insights, and the time and efforts spent reviewing our work. We consider each comment carefully below, but let us begin by addressing a common concern regarding the computational complexity. While the asymptotic complexity of the algorithms is high, there are some interesting special cases for which the algorithms are practical.

1. Our learning algorithm is able to infer the winning distributions of slates up to size $\sim k^2$ by looking at the $\sim n^k$ slates of size at most $k$. For moderate values of $k$, e.g., constant $k$, this algorithm is practical and is a significant improvement over the trivial $\sim n^{k^2}$ algorithm.

2. For modest values of $n$, e.g., learn the winning distributions over the $n$ restaurants in a town, our algorithm only requires $\sim 2^{\sqrt{n} \lg n}$ queries to approximate the full-slate, while the previous best algorithm needs $\sim 2^{n}$ queries.

In general, we view our results as the necessary first steps towards understanding the learnability of RUMs, which has been a long-standing research question.

Since the CFP welcomes “Theory” contributions in “learning theory”, we hope that the theoretical nature of our contribution will not be seen as a limitation.

---

### Decision · Program_Chairs · 2024-09-25

**Decision:**

Accept (poster)

**Comment:**

This paper has developed very strong results for the problem of learning RUMs. The improvement relative to previous work is enormous, improving the order of $n$ in the exponent itself, along with matching lower bounds. The analysis is technically sophisticated and interesting, connecting to boolean function analysis and polynomial approximation. This paper has very strong support from two reviewers and less support from two lower-confidence reviewers.  One criticism from several reviewers is the exponential complexity, but the authors have shown that this is unavoidable and hence I do not find that this should be viewed as a weakness. Another criticism that came out during the discussion period with the reviewers is that the paper uses an existing connection with boolean function analysis and polynomial approximation; however, I am not convinced that this is a valid criticism since (as noted by another reviewer) boolean function analysis and polynomial approximation are entire fields of study. It does seem that some heavy lifting is done by the prior work of Sherstov (2008), but finding these connections and the authors' own analysis looks impressive. This work is a welcome addition to the proceedings.